# 5′-Modifications improve potency and efficacy of DNA donors for precision genome editing

Krishna S Ghanta[1§], Zexiang Chen[1§], Aamir Mir[1†#], Gregoriy A Dokshin[1†¶], Pranathi M Krishnamurthy[1**], Yeonsoo Yoon[2], Judith Gallant[2‡], Ping Xu[2‡], Xiao-Ou Zhang[3], Ahmet Rasit Ozturk[1], Masahiro Shin[4], Feston Idrizi[4], Pengpeng Liu[4], Hassan Gneid[1,5], Alireza Edraki[1], Nathan D Lawson[4,6,7], Jaime A Rivera-Pérez[2,6], Erik J Sontheimer[1,6,7]*, Jonathan K Watts[1,5,6]*, Craig C Mello[1,7,8]*

[1]RNA Therapeutics Institute, University of Massachusetts Medical School, Worcester, United States; [2]Department of Pediatrics, Division of Genes and Development, University of Massachusetts Medical School, Worcester, United States; [3]Program in Bioinformatics and Integrative Biology, University of Massachusetts Medical School, Worcester, United States; [4]Department of Molecular, Cell and Cancer Biology, University of Massachusetts Medical School, Worcester, United States; [5]Department of Biochemistry and Molecular Biotechnology, University of Massachusetts Medical School, Worcester, United States; [6]Li Weibo Institute for Rare Diseases Research, University of Massachusetts Medical School, Worcester, United States; [7]Program in Molecular Medicine, University of Massachusetts Medical School, Worcester, United States; [8]Howard Hughes Medical Institute, University of Massachusetts Medical School, Worcester, United States

*For correspondence:
Erik.Sontheimer@umassmed.edu (EJS);
Jonathan.Watts@umassmed.edu (JKW);
Craig.Mello@umassmed.edu (CCM)

[†]These authors also contributed equally to this work
[‡]These authors also contributed equally to this work
[§]These authors also contributed equally to this work

Present address: [#]Tessera Therapeutics, Cambridge, United States; [¶]Vertex Pharmaceuticals, Boston, United States; [**]Bristol Myer Squibb, Cambridge, United States

**Abstract** Nuclease-directed genome editing is a powerful tool for investigating physiology and has great promise as a therapeutic approach to correct mutations that cause disease. In its most precise form, genome editing can use cellular homology-directed repair (HDR) pathways to insert information from an exogenously supplied DNA-repair template (donor) directly into a targeted genomic location. Unfortunately, particularly for long insertions, toxicity and delivery considerations associated with repair template DNA can limit HDR efficacy. Here, we explore chemical modifications to both double-stranded and single-stranded DNA-repair templates. We describe 5′-terminal modifications, including in its simplest form the incorporation of triethylene glycol (TEG) moieties, that consistently increase the frequency of precision editing in the germlines of three animal models (*Caenorhabditis elegans*, zebrafish, mice) and in cultured human cells.

## Introduction

Precision genome editing by homology-directed repair (HDR) often requires cells to use exogenously supplied DNA templates (donors) to repair targeted double-strand breaks (DSBs). Maximizing precision genome editing, therefore, requires understanding how cells respond both to DSBs and to exogenous donors. These responses can be influenced by many variables, including cell-intrinsic factors (e.g., genetics, cell type, and cell cycle stage) and cell-extrinsic factors (e.g., donor length, strandedness, and chemistry) (*Lin et al., 2014*; *Wienert et al., 2020*; *Nambiar et al., 2019*; *Renaud et al., 2016*; *Rees et al., 2019*; *Jayathilaka et al., 2008*; *Pinder et al., 2015*; *Riesenberg and*

*Maricic, 2018*; *Yu et al., 2015*; *Canny et al., 2018*; *Robert et al., 2015*). Each of these variables can influence the relative efficiency of HDR compared to competing DSB repair pathways, such as non-homologous end joining (NHEJ) (*Frank-Vaillant and Marcand, 2002*; *Mao et al., 2008*; *Chu et al., 2015*; *Maruyama et al., 2015*).

In many organisms and cell types, high HDR efficiencies are readily achieved using short single-stranded oligodeoxynucleotide (ssODN) donor templates that permit single base changes or short insertions or deletions. However, HDR is frequently less efficient when longer double-stranded DNA (dsDNA) templates are used as donors. It is not known why longer DNA donors yield lower rates of HDR. In many cell types, high concentrations of dsDNA cause cytotoxicity, limiting the number of long donor molecules that can be safely delivered into cells. In addition, due to their size, long donor molecules may not transit the nuclear envelope as efficiently, reducing the effective concentration at the site of repair, or requiring cell division to gain access to the target locus. Moreover, end-joining ligation reactions assemble linear dsDNA molecules into concatemers in eukaryotic cells (*Perucho et al., 1980*; *Folger et al., 1982*; *Mello et al., 1991*; *Stuart et al., 1988*; *Lacy et al., 1983*), further limiting the number of individual donor molecules and their ability to diffuse to their DSB target sites.

In an effort to improve nuclear delivery and HDR efficacy, we incorporated 5′-modifications into the donor molecules, including a simple triethylene glycol (TEG) moiety, a 2′-O-methyl (2′OMe) RNA::TEG modification, and a peptide nucleic acid (PNA) comprising the SV40 nuclear localization signal (NLS) (see Materials and methods) (*Brandén et al., 1999*). These 5′-modified donors increased the efficiency of templated repair by 2- to 5-fold in cultured mammalian cells as well as germline editing of *Caenorhabditis elegans*, zebrafish (*Danio rerio*) and mouse (*Mus musculus*). The modified donors exhibited a striking reduction in DNA ligation reactions including reduced self-ligation into concatemers and reduced sequence-independent ligation into cellular DSBs, suggesting that the 5′-modifications reduce the availability of 5′-ends for competing NHEJ reactions.

## Results
### End-modified DNA donors increase the efficiencies of HDR in mammalian cells

To examine the effects of donor end modifications on HDR in cultured mammalian cells, we took advantage of a modified traffic light reporter (TLR) comprising a 'broken' GFP coding region followed by a frameshifted mCherry coding region (*Certo et al., 2011*; *Iyer, 2019*). Cas9 targets the 'broken' GFP, which can only be made functional if precisely repaired by HDR, resulting in green fluorescence. If Cas9-mediated DSBs are imprecisely repaired by NHEJ, approximately one-third of the imprecise repair events will restore the reading frame of mCherry, resulting in red fluorescence. Cas9 and single guide RNA (sgRNA) expression vectors and dsDNA donors with or without 5′-modifications were electroporated into HEK293T TLR cells (*Figure 1A*), followed by flow cytometry to determine the percentage of cells expressing either GFP or mCherry.

We first examined the performance of dsDNA donors modified with 15 nucleotide (nt) 2′OMe-RNA fused to triethylene glycol (RNA::TEG). Strikingly, the frequency of HDR increased with the amount of RNA::TEG-modified donor to a maximal 52 % GFP+ cells at 1.2 pmol of donor before falling off at higher amounts of donor (*Figure 1B*). By contrast, a maximum HDR frequency of only 25 % GFP+ cells was observed at 1.6 pmol of unmodified donor. Notably, 0.4 pmol RNA::TEG-modified donor was as efficient as 1.6 pmol unmodified donor, suggesting that the modified donor is ~4 -fold more potent than the unmodified donor (*Figure 1B*). The increase in GFP+ cells was accompanied by a corresponding reduction in mCherry+ cells (*Figure 1C*).

We reasoned that the 2′OMe-RNA linker could be used to anneal PNA oligos attached to peptides that might enhance nuclear uptake. To test this idea, we produced complementary PNA oligos linked to a nuclear localization signal peptide or complementary PNA alone and tested these for HDR. Annealing these PNA oligos was well tolerated and did not diminish HDR, however neither did they enhance HDR (*Figure 1—figure supplement 1A-D*). Thus, further study will be needed to determine if RNA::TEG adapters can be used to append peptides or other molecules (e.g., CAS9 ribonucleoprotein [RNP]) that stimulate HDR.

We next used the TLR assay to define features of the RNA::TEG moiety that promote maximal HDR. Nucleofection of 1.2 pmol donors modified with 2′OMe-RNA, TEG, or covalent RNA::TEG moieties all

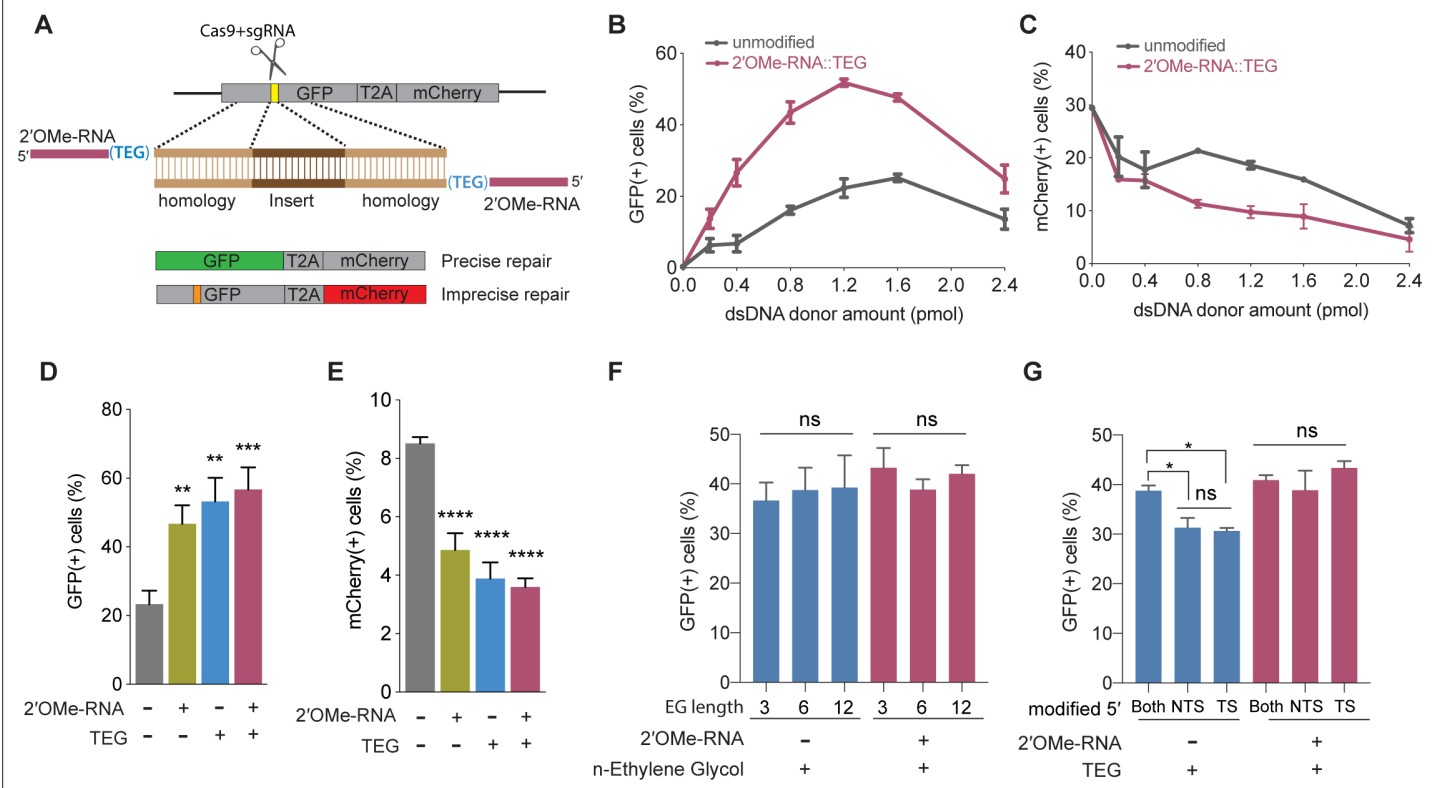

**Figure 1.** 5'-End-modified donors promote homology-directed repair (HDR) in traffic light reporter (TLR) cells. (**A**) Schematic showing the TLR assay to quantify HDR efficiencies using unmodified or end-modified double-stranded DNA (dsDNA) donors. Editing efficiencies plotted as percentage of (**B**) green fluorescent protein (GFP)+ (HDR) and (**C**) mCherry+ (non-homologous end joining [NHEJ]) HEK293T TLR cells at different amounts of unmodified, 2'-O-methyl (2'OMe)-RNA::triethylene glycol (TEG)-modified dsDNA donors. Editing efficiencies plotted as percentage of (**D**) GFP+ (HDR) and (**E**) mCherry+ (NHEJ) HEK293T TLR cells at 1.2 pmol of dsDNA donors indicated. Percentage of GFP+ cells obtained with dsDNA donors modified with various lengths of ethylene glycol (**F**) and with modifications to only one end or both 5'-ends of the donor. TS, target strand; NTS-,non-target strand (**G**). Mean ± s.d. for at least three independent replicates are plotted; two replicates for TEG donor in panel G.

The online version of this article includes the following figure supplement(s) for figure 1:

**Figure supplement 1.** 2'-O-methyl (2'OMe)-RNA at 5'-ends of donors promote homology-directed repair (HDR) in mammalian cells.

boosted HDR while reducing NHEJ events (*Figure 1D and E*). Increasing the length of the ethylene glycol moiety (3, 6, or 12 repeats) supported similar levels of HDR with or without the 2'OMe-RNA moiety (*Figure 1F*). Finally, donors with TEG modification at both 5'-ends yielded slightly better HDR efficiencies than donors with modification at only one of the two 5'-ends (*Figure 1G*). However, donors with RNA::TEG modification at both 5'-ends or at only one of the 5'-ends yielded similar HDR efficiencies (*Figure 1G*).

To explore the utility of TEG- and RNA::TEG-modified donors for repair at other genomic loci, we generated donors to integrate full-length *eGFP* at the endogenous *TOMM20*, *GAPDH*, and *SEC61B* loci (*Figure 2A*). We found that TEG or RNA::TEG donors consistently exhibited increased HDR levels in HEK293T cells as measured by the fraction of cells expressing eGFP at *TOMM20* (2-fold), at *GAPDH* (3-fold), and at SEC61B (5-fold) when compared to unmodified dsDNA donor (*Figure 2B–D*). RNA::TEG-modified donors also substantially increased HDR in two cell types that are less amenable to editing, increasing HDR at the *TOMM20* locus in human foreskin fibroblasts (HFF) cells (2.3-fold) and at the *Gapdh* locus in Chinese hamster ovary (CHO) cells (6-fold) (*Figure 2E–F*).

Next, to quantify the nature of repair outcomes (precise and imprecise), we employed deep sequencing assays. To facilitate sequencing across the repair site, we replaced a 12-nt sequence with a 9-nt sequence at the *EMX1* locus in HEK293T. We compared HDR efficiencies in this assay using unmodified, TEG-modified, and RNA::TEG-modified dsDNA donors with 90 base pair (bp) homology arms (*Figure 2G*). At 1.2 and 2.4 pmol, RNA::TEG-modified donors yielded 2-fold more precise edits compared to the unmodified donors. When even higher doses (5 pmol) were used, the gap in efficacy

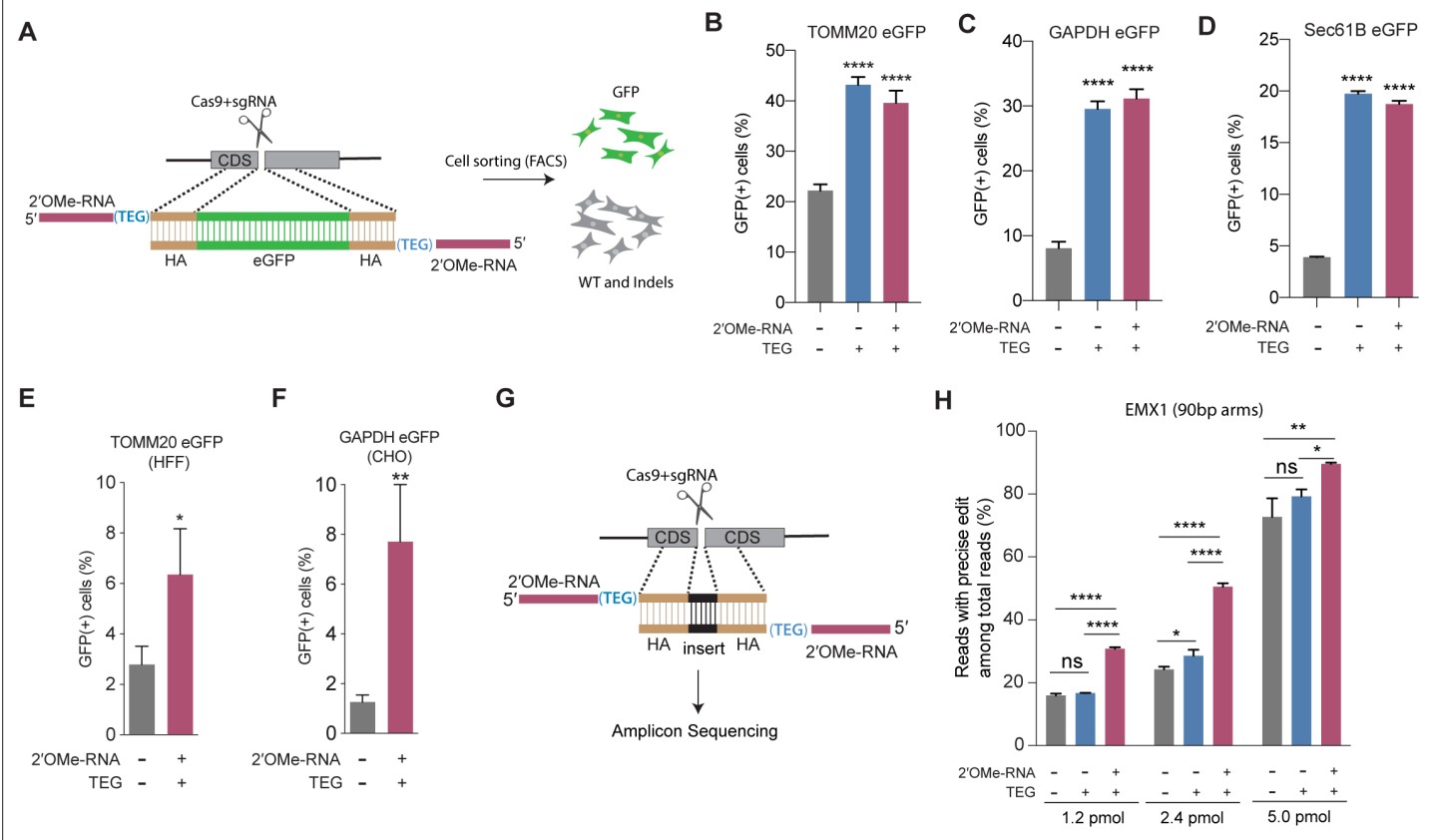

**Figure 2.** End-modified donors promote homology-directed repair (HDR) at endogenous loci in mammalian cell cultures. (**A**) Schematic representation of the 5′-modified donor design for enhanced green fluorescent protein (eGFP) insertion and fluorescence-activated cell sorting (FACS) is shown. Efficacy of eGFP integration at (**B**) *TOMM20* and (**C**) *GAPDH* (**D**) Sec61B loci in HEK293T cells using unmodified, triethylene glycol (TEG) or 2′-*O*-methyl (2′OMe)-RNA::triethylene glycol (TEG)-modified donors are plotted as percentage of GFP+ cells. Efficacy of eGFP integration at the (**E**) *TOMM20* locus in human foreskin fibroblast (HFF) (747 bp knock-in with ~1 kb homology arms) and (**F**) *Gapdh* locus in Chinese hamster ovary (CHO) (1635 bp knock-in with ~800 bp homology arms) cells using double-stranded DNA (dsDNA) (500 ng) donors with and without 2′OMe-RNA::TEG modifications at the 5′-ends. (**G**) Schematic representation of the dsDNA donor design used for quantification with deep sequencing is shown. (**H**) Illumina sequencing reads with precise knock-in are plotted for dsDNA donors with 90 bp homology arms at EMX1 locus in HEK293T cells. Mean ± s.d. for at least three independent replicates are plotted. p-Values were calculated using one-way ANOVA and in all cases end-modified donors were compared to unmodified donor unless indicated otherwise (Tukey's multiple comparisons test; ****p < 0.0001; ***p < 0.001; **p < 0.01; *p < 0.05; ns, not significant).

The online version of this article includes the following figure supplement(s) for figure 2:

**Figure supplement 1.** RNA::TEG (triethylene glycol) donors with short (90 bp) homology arms are more potent than unmodified donors at EMX1 locus.

between unmodified and RNA::TEG-modified donors narrowed to just 16 % ( 72.8% vs. 89.5%) precise reads (*Figure 2H*). The *EMX1* donor with 90 -bp homology arms also supported high levels of HDR in K562 cells across a broad dose range. Notably, low doses of donor supported higher levels of HDR in K562 cells than in HEK293T cells, suggesting that K562 cells are more susceptible to editing (*Figure 2—figure supplement 1*). In this assay, donors modified with TEG alone exhibited no benefit over unmodified donors (*Figure 2H* and *Figure 2—figure supplement 1*). Baseline HDR efficiencies obtained with unmodified donors vary from locus to locus. This may be caused due to the differences in cutting efficiencies, chromatin structure, and microhomology near the DSBs. These factors may also influence the fold change increase by end-modified donors.

## 5′-Modifications increase potency of single-stranded DNA donors

The experiments described thus far employed dsDNA donors; however, long single-stranded DNA (ssDNA) or short ssODN donors are also widely used in many HDR editing protocols. We therefore decided to explore how 5′-end modifications affect single-stranded donors of different lengths. Using the TLR assay, we found that addition of RNA::TEG at the 5′-end of a long (800 nt) ssDNA

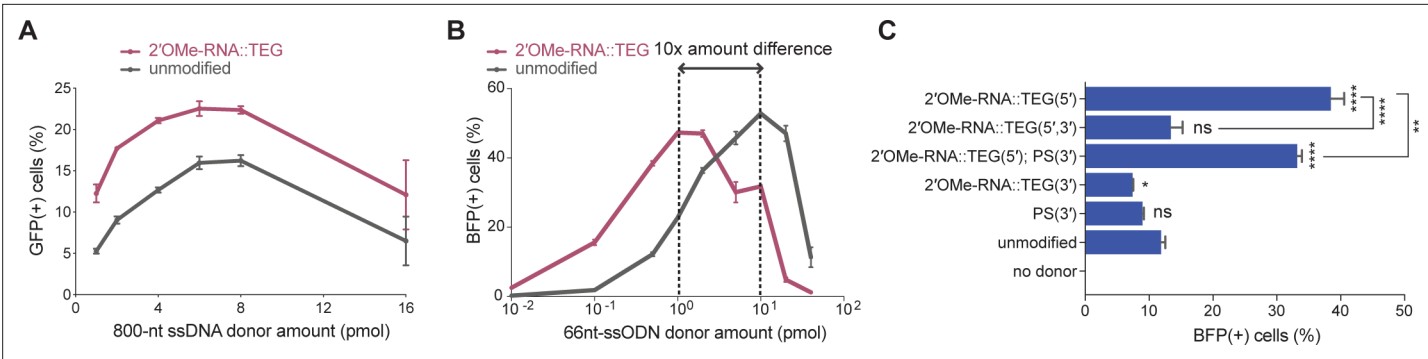

**Figure 3.** End modifications increase potency of single-stranded oligodeoxynucleotide (ssODN) donors. (**A**) Editing efficacy plotted as percentage of green fluorescent protein (GFP+) (precise) HEK293T traffic light reporter (TLR) cells at different amounts of unmodified and 2'-*O*-methyl (2'OMe)-RNA::triethylene glycol (TEG)-modified long single-stranded DNA (ssDNA) donors (800 nt). (**B**) Editing efficacy of GFP-to-blue fluorescent protein (BFP) reporter conversion in K562 cells using different amounts of unmodified and 2'OMe-RNA::TEG-modified 66 nt single-stranded oligodeoxynucleotide (ssODN) donors plotted as percentage of BFP+ cells (homology-directed repair [HDR]). (**C**). Editing efficacy of GFP-to-BFP conversion in K562 cells using 0.5 pmol of ssODN donors modified at the 5'-end alone, the 3'-end alone, or at both the 5'- and 3'-ends, with phosphorothioate (PS), TEG, 2'OMe-RNA, or 2'OMe-RNA::TEG, plotted as percentage of BFP+ cells (precise). Complete figure of panel C is shown, along with other modifications, in *Figure 3—figure supplement 2*. Mean ± s.d. for at least three independent replicates are plotted. p-Values were calculated using one-way ANOVA and in all cases end-modified donors were compared to unmodified donor unless indicated otherwise (Tukey's multiple comparisons test; ****p < 0.0001; ***p < 0.001; **p < 0.01; *p < 0.05; ns, not significant).

The online version of this article includes the following figure supplement(s) for figure 3:

**Figure supplement 1.** 2'-*O*-methyl (2'OMe)-RNA::triethylene glycol (TEG) modification of single-stranded DNA (ssDNA) donors results in reduced imprecise editing.

**Figure supplement 2.** Effects of terminal and non-terminal modifications of single-stranded oligodeoxynucleotide (ssODN) donors on homology-directed repair (HDR) efficacy.

donor significantly boosted HDR compared to the unmodified ssDNA donor. The frequency of HDR increased with the dose of ssDNA donor, reaching maximal HDR (22.5 % GFP(+)cells) at 6–8 pmol donor amounts (*Figure 3A*, *Figure 3—figure supplement 1A*). The RNA::TEG-modified donor was greater than 4-fold more potent than the unmodified donor reaching a threshold of 16 % GFP(+) cells at a concentration of ~2 pmol whereas achieving the same threshold of 16 % required 8 pmol of unmodified donor (*Figure 3A*).

High yields of HDR in cultured mammalian cells have been achieved using short synthetic ssODN donors (*Richardson et al., 2016*). To test 5'-modified ssODNs for HDR efficacy, we used a sensitive green-to-blue fluorescent protein (GFP-to-BFP) conversion assay in K562 cells. Precise editing converts a functional GFP sequence to BFP sequence, producing cells that are GFP(-) and BFP(+). Imprecise editing produces cells that are both GFP(-) and BFP(-) (*Glaser et al., 2016*). Using 66 nt long ssODN donors and titrating the amount over a range of 0.01–40 pmol, we found that RNA::TEG and unmodified donors produced similar maximal levels of HDR (47.5–52.8% BFP(+) cells). However, maximal HDR required 10-fold less RNA::TEG-modified ssODN than unmodified donors (*Figure 3B*). We also observed reduced levels of imprecise editing (GFP(-) and BFP(-)) as the frequency of HDR increased (*Figure 3—figure supplement 1B*). For both donor types, the decline in editing at higher doses correlated with the appearance of dead cells (data not shown), suggesting that dose-limiting toxicity scales with increased HDR potency.

The use of fully synthetic ssODN donors allowed us to explore additional modifications, including internal and 3'-modifications. Interestingly, 2'OMe-RNA, RNA::TEG, or TEG moieties at the 3'-terminus did not enhance HDR compared to unmodified ssODN, and moreover, they blocked the ability of 2'OMe-RNA, RNA::TEG, or TEG moieties at the 5'-end to enhance HDR (*Figure 3C*, *Figure 3—figure supplement 2*). By contrast, HDR was neither enhanced nor impeded by phosphorothioate (PS) linkages placed at 5'- or 3'-terminal linkages at the doses tested (*Figure 3—figure supplement 2*). Taken together these findings suggest that the mechanism of HDR improvement requires an available 3'-OH.

## 5′-Modified donors promote precision germline editing in *C. elegans*

Efficient genome editing in *C. elegans* can be achieved by directly injecting mixtures of Cas9 RNP complex and donor into the syncytial ovary (*Cho et al., 2013*; *Paix et al., 2015*; *Dokshin et al., 2018*), producing dozens of independent precision editing events among the progeny of each injected animal (*Ghanta and Mello, 2020*). We designed unmodified, TEG-modified, and RNA::TEG-modified donors to insert *gfp* at the *csr-1* locus or to correct *eft-3p::gfp* reporter that contains partial sequence of *gfp* (see Materials and methods; *Figure 4A*). To monitor injection quality, we co-injected a plasmid encoding the transformation marker *rol-6(su1006)*, which produces the Roller phenotype. The TEG- and RNA::TEG-modified donors produced about twice as many GFP(+) progeny per injected animal than did the unmodified donor (*Figure 4B and E*, two representative broods per donor). Among the Roller cohort, which was previously shown to exhibit lower editing efficiency (*Ghanta and Mello, 2020*), end-modified donors increased the fraction of GFP(+) Roller progeny by several fold. For example, whereas the unmodified *eft-3* donor produced only 12.6 % GFP-positive Rollers, the TEG- and RNA::TEG-modified *eft-3* donors produced 57.1% and 49% GFP-positive Rollers (*Figure 4C*). Similarly, GFP::CSR-1(+) Rollers increased from 8.8 % (unmodified) to 28 % (TEG) and 32.8 % (RNA::TEG) (*Figure 4F*). TEG- and RNA::TEG-modified *eft-3* and *csr-1* donors produced >50% GFP(+) non-Roller progeny compared to roughly 22 % (*eft-3*) and 30 % (*csr-1*) GFP(+) non-Rollers produced by the unmodified donors (*Figure 4D and G*). Every GFP(+) animal tested transmitted the edit to the next generation (*Figure 4—figure supplement 1*). Thus, compared to the unmodified donors, the 5′-TEG and 5′-RNA::TEG donors substantially increase the frequency of *gfp* insertion by HDR in the *C. elegans* germline. Strikingly, end-modified donors frequently yielded more than 100 independent GFP(+) F1 progeny from a single injected hermaphrodite.

## 5′-Modified donors promote precision editing in vertebrate zygotes

We next asked if donor 5′-modifications improve precision genome editing in zebrafish and mouse zygotes. For zebrafish genome editing, we designed 147 bp dsDNA donors to insert the 45 nt Avitag sequence into the 5′-end of the *Hey2* coding sequence (*Figure 5A*). Unmodified or end-modified donors were co-injected with Cas12a RNPs into one-cell embryos (see Materials and methods), and editing efficiencies were quantified by high-throughput sequencing using genomic DNA isolated 24 hr after injection (*Liu et al., 2019*). Strikingly, the frequency of precise editing was 11-fold higher with the RNA::TEG (4.4%) donor than with the unmodified donor (0.4%) (*Figure 5A*). The TEG-modified donor however failed to enhance precise editing in zebrafish zygotes (*Figure 5A*). The total level of editing was comparable in each condition as shown by the fraction of reads with indels (*Figure 5—figure supplement 1*).

To test whether RNA::TEG-modified donors enhance precise editing in mouse zygotes, we targeted the *Tyrosinase (Tyr)* and *Sox2* loci. First, we sought to convert the coat color of Swiss-Webster albino (*Tyr^c*) mice to a pigmented phenotype (*Tyr^c-cor*; cor: corrected) using a donor to replace the serine 103 codon (TCT) with a cysteine (TGC) codon. The donor also introduces six silent mutations to prevent the guide RNA from directing cleavage of the edited locus (*Figure 5B*). We injected unmodified or RNA::TEG-modified donors with Cas9 RNPs into zygotes, transferred the embryos into pseudo-pregnant females, and quantified the repair efficiency by phenotyping the coat color of founder (F0) mice. The RNA::TEG-modified donor yielded more than twice as many pigmented F0 mice (37.9 % uniform or mosaic) compared to unmodified donor (17.4%) (*Figure 5B*, *Figure 5—figure supplement 2A*). Strikingly, most (92%) of the edited founders produced by the RNA::TEG-modified donor had uniformly pigmented coats, whereas only 62.5 % of the edited F0 produced by the unmodified donor had a uniformly pigmented coat color (*Figure 5C*; *Figure 5—figure supplement 2A*), suggesting that the RNA::TEG-modified donor promotes editing during early zygotic divisions. Representative images of F0 litters with dark coat color are shown in *Figure 5D*. We confirmed that F0 mice with pigmented coat transmitted the corrected *Tyr^c-cor* allele to F1 pups (*Figure 5—figure supplement 2B, C*). Taken together, these results show that RNA::TEG donors are at least 2-fold more efficient than unmodified donors in mouse zygote editing.

Next, we sought to insert a sequence encoding an in-frame V5 epitope immediately before the stop codon at the 3′-end of the *Sox2* locus (*Figure 5E*). We injected unmodified or RNA::TEG-modified donors with Cas9 RNPs into zygotes, transferred the embryos into pseudo-pregnant mice, and geno-typed F0 progeny by PCR across the *Sox2* target site and Sanger sequencing. The V5 tag was precisely

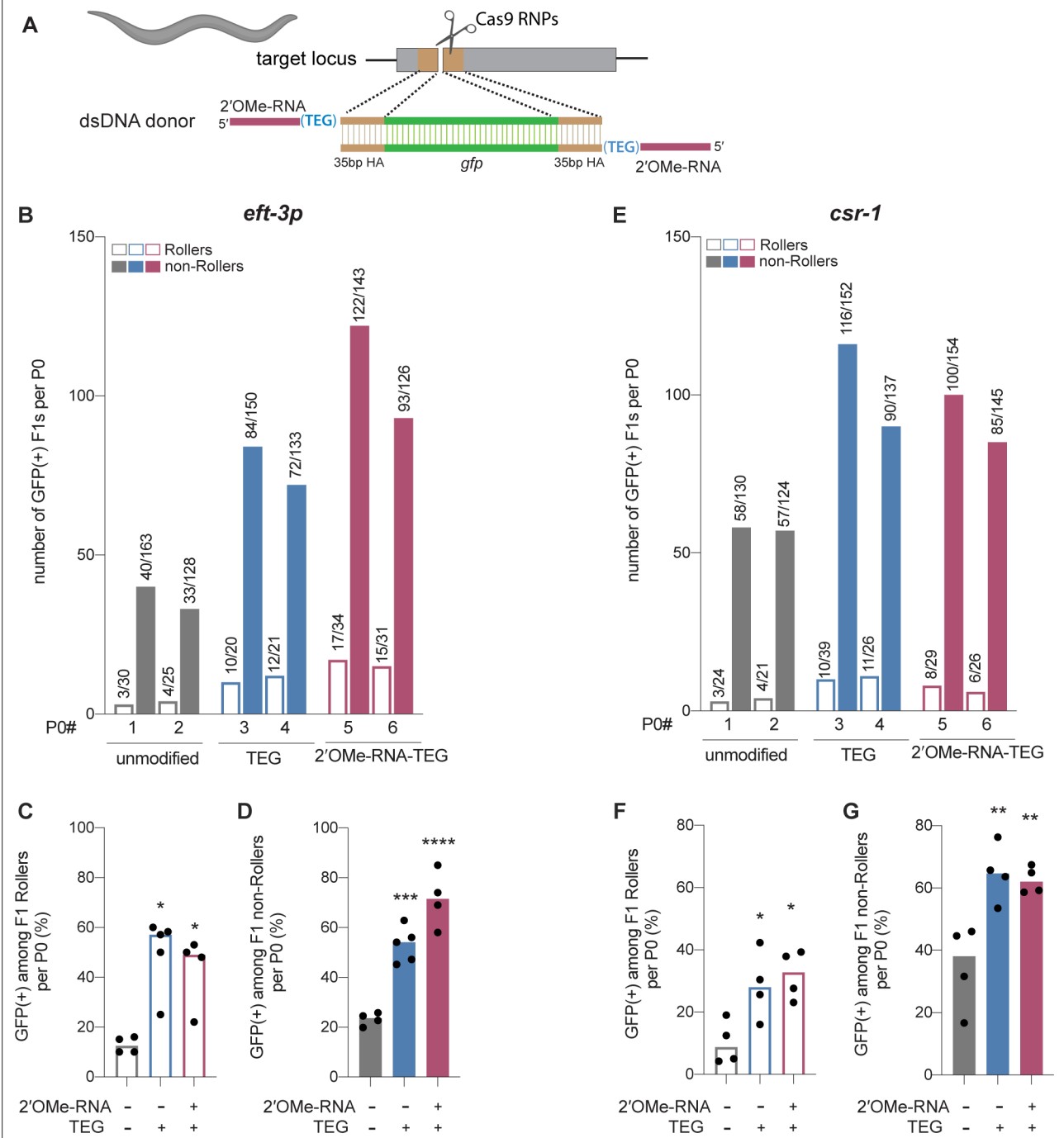

**Figure 4.** Modified donors promote precise editing in *Caenorhabditis elegans*. (**A**) Schematic showing end-modified double-stranded DNA (dsDNA) donors (25 ng/µl) with short (~35 bp) homology arms to insert *gfp* tag. (**B**) Number of green fluorescent protein (GFP) expressing animals among entire F1 brood of two representative P0 animals for each donor type are plotted for *eft-3p* reporter locus. Fraction of F1 animals expressing GFP among (**C**) Roller and (**D**) non-Roller cohorts are plotted as percentage for *eft-3p* locus. Similarly, (**E**) number of GFP expressing animals among two representative broods, fraction of F1 animals expressing GFP among (**F**) Roller and (**G**) non-Roller cohorts are plotted for *csr-1* locus. Open bars (Rollers) and closed bars represent (non-Rollers) median. Number of GFP expressing animals among total number of animals scored per cohort are shown above the bars. n ≥ 4 broods for each donor condition. p-Values were calculated using one-way ANOVA and in all cases end-modified donors were compared to unmodified donors (Tukey's multiple comparisons test; ****$p < 0.0001$; ***$p < 0.001$; **$p < 0.01$; *$p < 0.05$; ns, not significant).

The online version of this article includes the following figure supplement(s) for figure 4:

**Figure supplement 1.** Precise insertions are germline transmitted in *Caenorhabditis elegans*.

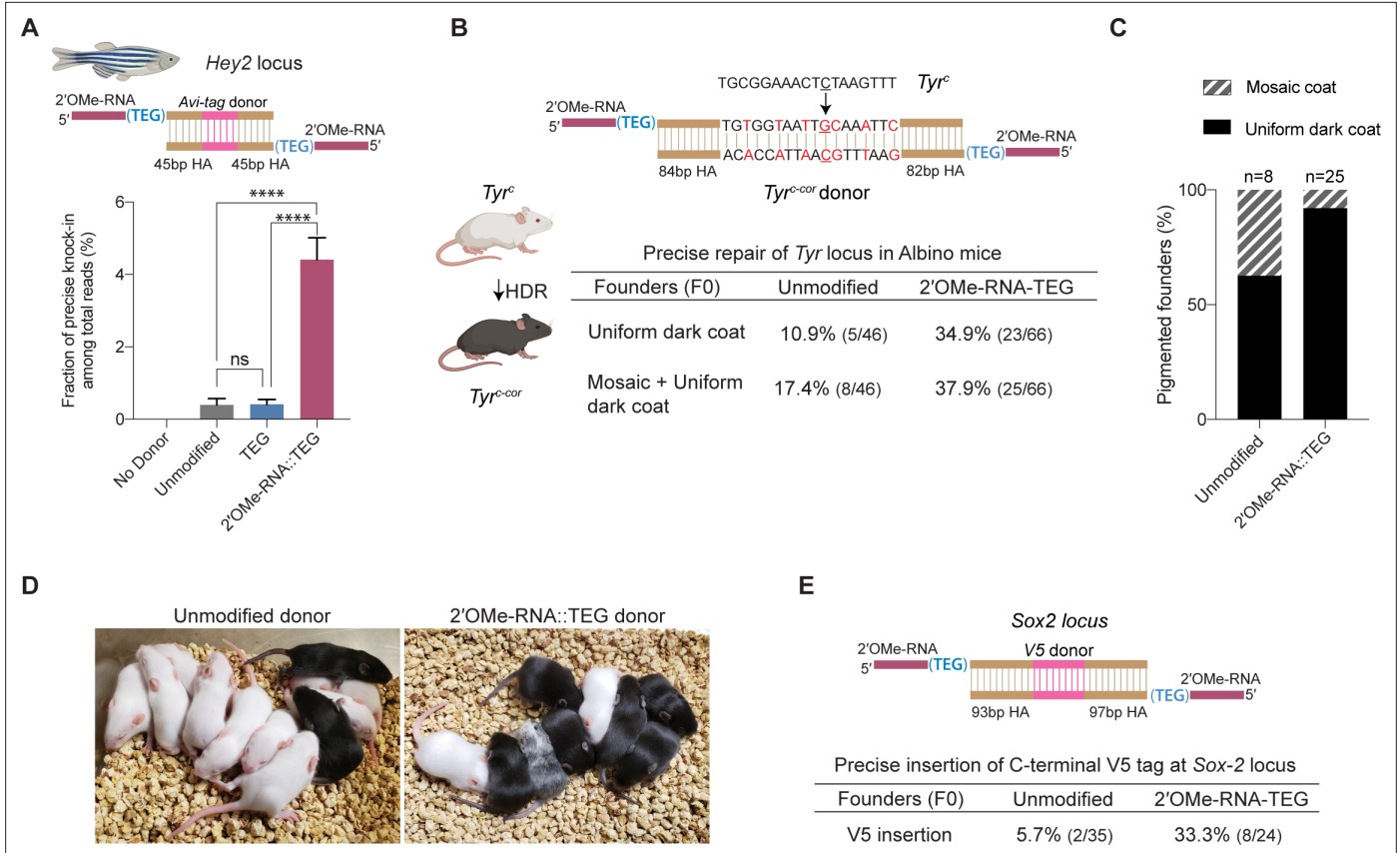

**Figure 5.** 2'-O-methyl (2'-OMe)-RNA-triethylene glycol (TEG) donors promote precise editing in vertebrate zygotes. (**A**) Unmodified, TEG, and 2'-OMe-RNA::TEG-modified double-stranded DNA (dsDNA) donors were injected into zebrafish zygotes. dsDNA donor design to knock-in Avi-tag is shown on the top and the fraction of Illumina reads containing precise knock-in are plotted as percentages. Mean ± s.d. for at least three independent replicates (two for unmodified donors) are plotted (**B**). Design of the dsDNA donors injected into mouse zygotes to precisely convert the coat color of albino mice ($Tyr^C$) to pigmented ($Tyr^{C-Cor}$) by editing C to G (underscored) along with six silent mutations (in red) is shown. Percentages of F0 founder mice with black coat are shown. (**C**) Percentages of animals among homology-directed repair (HDR)-positive F0s that have uniform dark coat or mosaic coat color are plotted for unmodified and 5'-modified donors. (**D**) Representative pictures of 10 -day- old F0 mice with pigmented (HDR) or white (wild-type [WT] or indel) coat color are shown. One mosaic mouse (third from left) can be seen among the pups obtained with end-modified donor. (**E**) Donor design to knock-in V5 tag at the C-terminus of Sox2 is shown on the top. Percentage of founder animals containing perfect $V5$ insertion at $Sox2$ locus are shown for each donor type. HA, homology arms. p-Values were calculated using one-way ANOVA (Tukey's multiple comparisons test; ****p < 0.0001; ***p < 0.001; **p < 0.01; *p < 0.05; ns, not significant).

The online version of this article includes the following figure supplement(s) for figure 5:

**Figure supplement 1.** Indel efficiencies in zebrafish zygotes.

**Figure supplement 2.** 5'-Modified donors improve targeted editing efficiency at the $Tyr$ locus in albino mice.

**Figure supplement 3.** 5'-Modified donors improve targeted editing efficiency at the $Sox2$ locus in mouse zygotes.

inserted into the $Sox2$ locus in only 5.7 % (n = 35) of F0 animals from the injection with unmodified donor. By contrast, the RNA::TEG-modified donor resulted in precise insertion of V5 in 33.3 % (n = 24) of the F0 animals—a greater than 5-fold increase in precise editing (**Figure 5E** and **Figure 5—figure supplement 3A**). All of the V5-positive founders tested (one F0 from the unmodified donor and six F0s from RNA::TEG-modified donor) transmitted the $Sox2::V5$ allele to F1 progeny and the insertion was confirmed by Sanger sequencing (**Figure 5—figure supplement 3B, C**). Thus, the 5'-RNA::TEG modification greatly improves the efficiency of precise genome editing in vertebrate model systems.

## 5'-Modifications suppress donor concatenation

Upon delivery into animal cells or embryos, linear DNA molecules are known to form extensive homology-mediated and ligation-dependent concatemers (**Figure 6A**; **Perucho et al., 1980**; **Folger et al., 1982**; **Mello et al., 1991**). We reasoned that 5'-modifications to the donor might suppress

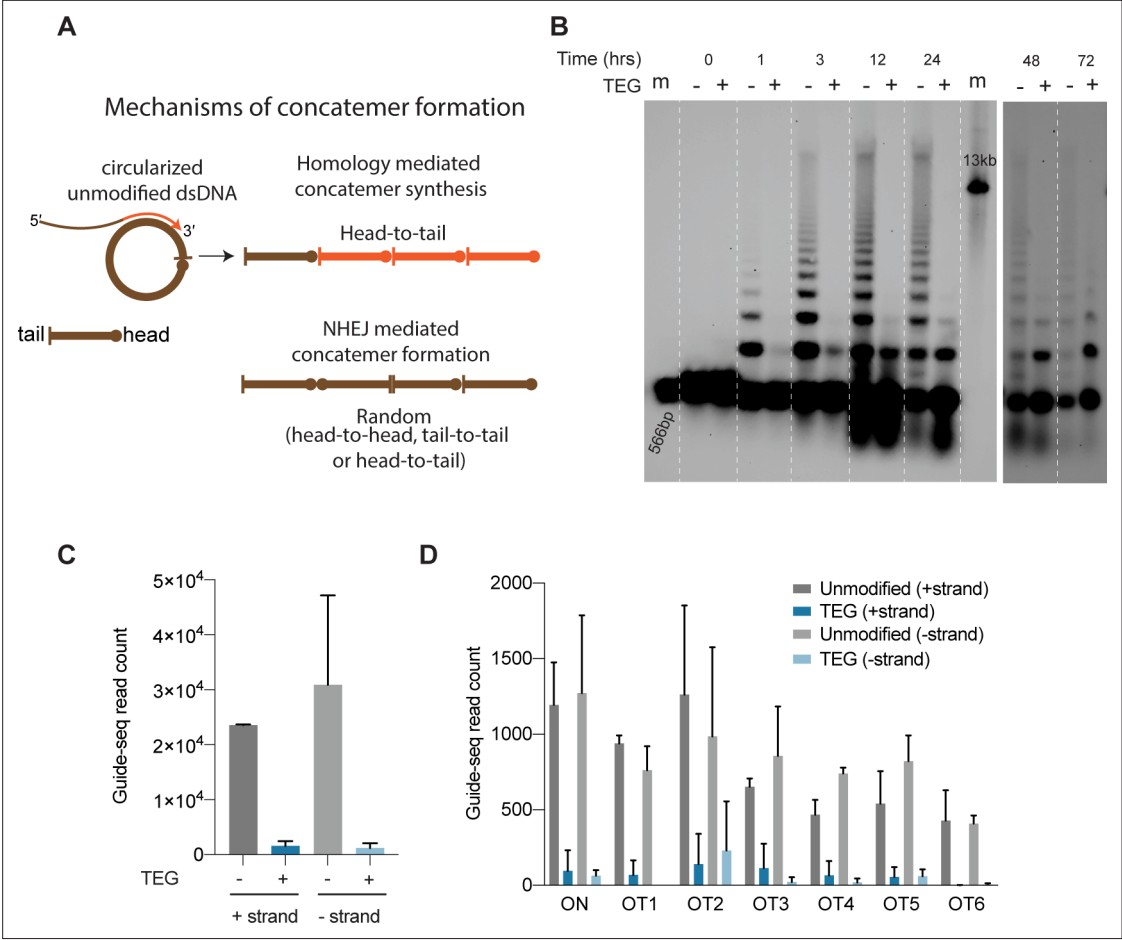

**Figure 6.** 5'-Modifications suppress donor ligation reactions. (**A**) Model for mechanisms of concatemer formation for unmodified donors is shown. (**B**) Southern blot of unmodified and triethylene glycol (TEG)-modified double-stranded DNA (dsDNA) (566 bp) nucleofected into HEK293T cells and collected at indicated time points. Concatemerization of unmodified DNA is visualized as ladders; 566 bp DNA and 13 kb long DNA are used as size markers (m). Number of GUIDE-seq reads with unmodified and TEG-modified short dsDNA (34 bp) integration for (**C**) whole genome and (**D**) on-target (*ARHGEF9*) and six previously validated off-target loci are plotted. Data from two biological replicates is shown.

The online version of this article includes the following figure supplement(s) for figure 6:

**Source data 1.** Uncropped full blot image of *Figure 6B* (0–24 hr).

**Source data 2.** Uncropped full blot image of *Figure 6B* (48–72 hr).

the formation of concatemers, thereby making linear donors more available for HDR. To test this idea, we nucleofected 566 bp dsDNA donors into HEK293T cells, harvested cells over a course of 3 days, and assessed the formation of concatemers by Southern blot analysis. We found that the unmodified dsDNA formed concatemers within 1 hr after nucleofection. These concatemers were composed of two to several copies of the DNA, inferred from the presence of a ladder of bands on the Southern blot (*Figure 6B*, *Figure 6—source data 1*). Concatemers of up to 10 copies were present within 3 hr after nucleofection and peaked in abundance by 12 hr. Concatemer levels declined over the next 12 hr but persisted at low levels until at least 72 hr after nucleofection. By contrast, the TEG-modified DNA showed a marked delay in the formation and levels of multimers (*Figure 6B*, *Figure 6—source data 2*). Dimers and trimers gradually formed over the first 12–24 hr but were present at much lower levels than those formed by unmodified DNA. At late time points—24, 48, and 72 hr after transfection—we observed a greater fraction of TEG-modified DNA monomers than unmodified monomers (*Figure 6B*). These results suggest that the 5'-TEG modification suppresses concatemer formation.

## End-modifications suppress direct ligation of short DNA into DSBs

To determine if TEG modification suppresses the direct ligation of TEG-modified linear molecules into chromosomal DSBs, we performed GUIDE-seq analyses (*Tsai et al., 2015*), which measures the incorporation of short (34-nt) dsDNA into on-target and off-target DSBs. Using the previously described dsDNA probe either with or without the TEG modifications, we targeted the *ARHGEF9* locus that was previously characterized for off-target editing (*Amrani et al., 2018*). Strikingly, the TEG-modified DNA produced 19-fold fewer GUIDE-seq reads (genome wide) than did the unmodified DNA (*Figure 6C*). The number of TEG-modified DNA insertions obtained at the on-target cut site in the *ARHGEF9* locus and at the top 6 off-target sites was dramatically reduced, ranging from 15-fold to 6-fold lower compared to insertions of the unmodified DNA (*Figure 6D*). Taken together these data suggest that TEG modifications suppress direct ligation of donor molecules both to each other and to chromosomal DSBs.

## Discussion

Here, we have explored how several types of chemical modifications to the repair template DNA affect the efficiency of precise homology-dependent repair. In mammalian cells, donors containing simple modifications such as TEG or 2'OMe-RNA::TEG on their 5'-ends improved HDR efficacy. These modifications increased the potency of ssDNA and dsDNA (long and short) donors, allowing efficient editing at significantly lower amounts. Modifying the ends of the donors suppressed concatemer formation and reduced random integration of short dsDNA at chromosomal DSBs.

End modifications affected long and short donors differently in mammalian cells. On long donors end modification caused a ~2- to 5-fold increase in HDR frequency (total efficacy) compared to unmodified donors and did so without changing the donor concentration where efficacy reached its plateau. In contrast, on short donors end modifications did not increase the maximal efficacy of HDR, but instead dramatically reduced the amount of donor required to reach that maximal level. Put another way, long DNA donors exhibited both increased potency and maximal efficacy when modified, while short ssODN and dsDNA donors exhibited increased potency but no increase in maximal efficacy. This difference requires further study but could be explained if shorter donors and longer DNA donors experience different dose-limiting barriers. For example, the dose-limiting toxicity of ssODNs could be driven by total number of free DNA ends per cell, while longer molecules could encounter dose-limiting toxicity driven by total DNA mass. Consistent with this idea, unmodified long dsDNA donors begin to plateau in efficacy at nearly 4-fold more mass, but ~10 -fold lower molar amounts than ssODNs. When end-modified, both types of donor exhibit similar maximal efficacy in the 1–2 pmol range.

RNA::TEG-modified donors significantly increased the levels of precision editing in three different model organisms (*C. elegans*, zebrafish, and mice). In all three animals, high HDR efficiencies were achieved using end-modified dsDNA donors, which in some cases approached efficiencies previously observed for ssODN donors (*Dokshin et al., 2018*; *Kan et al., 2017*). Importantly, precise insertions were obtained with relatively short homology arms. For example, in mouse zygote injections, we used donors with homology arms of less than 90 bp, similar to typical arm lengths used for ssODN donors (*Quadros et al., 2017*) and at relatively low concentrations (1 ng/µl).

How do end modifications help increase the efficacy of the donors? Our findings suggest that they do so, in part, by suppressing non-homologous end-joining reactions. In several systems dsDNA donors have been shown to quickly form extrachromosomal arrays (*Perucho et al., 1980*; *Folger et al., 1982*; *Mello et al., 1991*) and may do so directly in the cytoplasm (*Forbes et al., 1983*). For example, DNA delivered into the cytoplasm of the *C. elegans* gonadal syncytium gains entry into oocytes over a 24 hr period in a manner more consistent with cytoplasmic flow than with direct nuclear uptake by germ nuclei (*Ghanta and Mello, 2020*), and transformants established in this way have been shown to contain concatenated arrays of injected DNA, several hundred kilobases in length, which then partition to progeny in a non-Mendelian fashion as extrachromosomal elements (*Mello et al., 1991*; *Stinchcomb et al., 1985*). Integration of similar extrachromosomal arrays into the host genome have been reported in zebrafish and mouse zygotes (*Stuart et al., 1988*; *Lacy et al., 1983*; *Costantini and Lacy, 1981*). Thus, the suppression of donor concatemer formation by 5'-modified donors could increase the effective molar amounts of donor available for precise repair

of the target DSB. Similarly, once in the nucleus, the suppression of direct ligation to chromosomal DNA through end-joining reactions could further increase precision repair. Perhaps consistent with suppression of concatenation as a major mechanism of action, it is intriguing that modification of a single end was nearly as effective as modifications to both ends of the donor. In principle, a single end modification would limit concatenation to dimer formation. Similarly, modification of a single end could prevent donors from ligating into circles which might then concatenate further through HDR.

In addition to increasing the amount of available donor molecules, another possible benefit of suppressing end-joining reactions is that the free ends of the donor might then be available to participate in the HDR mechanism (e.g., by assembling elements of the DSB repair machinery directly on the free 3′-end of the donor). We found that a free unmodified 3′-end was required for efficient HDR. Thus, by suppressing ligation, the 5′-modification in effect maintains available 3′-ends, perhaps to prime repair synthesis.

In previous studies, fluorescent and amine modifications to the 5′- and 3′-termini of ssODN donors did not improve HDR efficacy over unmodified donors (*Lee et al., 2017*). Similarly, PS linkages were shown to improve HDR (*Renaud et al., 2016*). However, these studies were performed using much higher concentrations than the optimal concentrations for modified donors determined here. In our study, ssODNs with PS linkages did not improve HDR at doses where RNA::TEG- and TEG-modified donors were most efficacious. While our study was in preparation (*Ghanta, 2018*), three studies explored donors with 5′-end modifications. One study showed that the addition of biotin improved HDR and favored single copy insertion in the rice fish medaka (*Gutierrez-Triana et al., 2018*). The biotin moiety was attached to the donor via a polyethylene glycol (PEG) linker, but the study did not explore donors with PEG alone. *Yu et al., 2020*, showed that PEG10 with a 6-carbon linker boosted precise GFP insertions in vertebrate cells similar to those reported here for TEG- and RNA::TEG-modified donors, and at similar concentrations to those we employed (*Yu et al., 2020*). The third study describes the suppression of NHEJ-mediated insertions using donors with 5′-biotin::PEG or 5′-ssDNA::PEG moieties in non-transformed cells (*Canaj, 2019*). Our studies are in agreement with these findings and extend them to additional modifications and to in vivo genome-editing applications in three animal systems.

Explorations of how modifications to the donor, both chemical and physical in nature, might alter HDR efficacy are far from complete. For example, a physical pre-treatment to the donor that enhanced HDR was discovered serendipitously when attempting to anneal oligonucleotides to the ends of a PCR donor. The act of simply melting and cooling long dsDNA PCR donors (~1 kb in length) improved HDR efficacy in *C. elegans* by an order of magnitude (*Ghanta and Mello, 2020*). Preliminary findings suggest that melting an 800 bp donor did not improve HDR efficiency in human cells (data not shown) but further studies are required to more rigorously test the impact of melting donors of greater length, and directed at more target sites, and of course to explore other physical perturbations.

The chemistry space for donor modification is also vast and remains largely unexplored. We do not know why donors modified with TEG and RNA::TEG performed similarly in *C. elegans*, while RNA::TEG was consistently superior to TEG alone in zebrafish and human cells. The *C. elegans* system is unique in that it targets meiotic pachytene nuclei. Perhaps TEG alone is sufficient to stabilize donor molecules for HDR in meiotic cells that are actively engaged in DNA recombination, while increased stability is required to enable donors to persist longer and thus to engage a much less active HDR repair machinery in mitotic cells. The RNA::TEG modification might therefore facilitate editing in mitotic cells by providing this measure of improved protection from nuclease activity compared to TEG alone. PS linkages are known to protect against nuclease activity (*Renaud et al., 2016*), and it will therefore be interesting to explore whether a combination of internal (e.g., PS linkages) and terminal (e.g., 5′-RNA::TEG or 5′-TEG) modifications can further increase HDR efficacy. Indeed, our results should incite the search for additional chemistries that could boost donor stability while still allowing the donor to serve as a template for repair polymerases; some such studies are underway in our laboratories. Future studies will also need to explore whether the incorporation of donor chemistries will synergize with other methods that stimulate HDR (*Lin et al., 2014*; *Wienert et al., 2020*; *Chu et al., 2015*; *Maruyama et al., 2015*; *Gutschner et al., 2016*; *Yang et al., 2016*; *Ling et al., 2020*).

## Materials and methods

### Synthesis of PNA-NLS peptide

PNA oligomers were synthesized at 2 µmol scale on Fmoc-PAL-PEG-PS solid support (Applied Biosystems) using an Expedite 8909 synthesizer. Fmoc/Bhoc-protected PNA monomers (Link Technologies) were dissolved to 0.2 M in anhydrous *N*-methylpyrrolidinone. Amino acid monomers (Sigma Aldrich) and AEEA linker (Link Technologies) were dissolved to 0.2 M in anhydrous dimethylformamide. Coupling time was 8.5 min using HATU (Alfa Aesar) as activator; double coupling was performed on all PNA monomers and amino acids. PNAs were cleaved and deprotected by treating the resin with 400 µl of 19:1 TFA:*m*-Cresol for 90 min at room temperature. The resin was then removed with a PTFE centrifugal filter and PNAs were precipitated from cold diethyl ether and resuspended in deionized water. PNAs were purified by HPLC on a Waters XSelect CSH C18 5 µm column at 60 °C, using gradients of acetonitrile in water containing 0.1 % TFA, and were characterized on an Agilent 6530 Q-TOF LC/MS system with electrospray ionization. The PNA::NLS sequence used was GCGCTCGG-CCCTTCC-[AEEA linker]-PKKKRK.

### Synthesis of PEGylated oligos

PEG-modified oligonucleotides were synthesized using standard phosphoramidite methods on an ABI 394 synthesizer. Phosphoramidites were purchased from ChemGenes. Coupling times for 2′OMe-RNA and spacer phosphoramidites were extended to 5 min. Oligonucleotides were deprotected in concentrated aqueous ammonia at 55 °C for 16 hr. Oligonucleotides were desalted using either Nap-10 (Sephadex) columns or Amicon ultrafiltration. All the PEG-modified oligonucleotides were characterized on an Agilent 6530 Q-TOF LC/MS system with electrospray ionization. The 2′-OMe RNA sequence appended to the 5′-end of donor DNAs was GGAAGGGCCGAGCGC.

PEGylated oligos can also be purchased from commercial sources such as Integrated DNA Technologies (IDT) at 100 nmol scale with simple desalting.

### dsDNA donor generation

Donor template sequences with the homology arms and the desired insert for knock-in (eg: gfp) were generated by PCR. PCR products were cloned into ZeroBlunt TOPO vector (Invitrogen, #450245) and plasmids were purified using Macherey-Nagel midi-prep kits (cat# 740412.50). Using the purified plasmids as templates and PEGylated oligos as primers, donor sequences were PCR-amplified with Q5 (NEB, *C. elegans*) or Q5 or Phusion polymerase (NEB, mammalian). Before use in *C. elegans* microinjections, the resulting PEGylated PCR products were excised from 0.8% to 1% TAE agarose gel and purified using spin columns (Omega, #D2501-02). For use in mammalian cells, the PEGylated long PCR products were purified using spin columns (Qiagen, #28104) and short PCR products were gel-extracted (Omega, #D2501-02) and then purified again with Ampure XP beads.

### ssDNA donor generation

Long ssDNA donors were prepared using the protocol described by *Li et al., 2019*. Briefly, the donor template containing the T7 promoter was amplified using standard PCR and purified using SPRI magnetic beads (Core Genomics). T7 in vitro transcription was performed using the HiScribe T7 High Yield RNA Synthesis kit (NEB) and the RNA was purified using the SPRI magnetic beads. Finally, the ssDNA donor was synthesized by TGIRT-III (InGex)-based reverse transcription using the synthesized RNA as a template and a TEG-modified or -unmodified DNA primer. We then performed base treatment to remove RNA. The donor was again purified using SPRI beads.

### Expression and purification of SpyCas9

The pMCSG7 vector containing the 6xHis-tagged 3xNLS SpyCas9 was a gift from Scot Wolfe at UMass Medical School. This construct was transformed into the Rosetta 2 DE3 strain of *Escherichia coli* for protein production. Expression and purification of SpyCas9 was performed as described previously (*Jinek et al., 2012*). Briefly, cells were grown at 37 °C to OD600 of 0.6, at which point 1 mM IPTG (Sigma) was added and the temperature was lowered to 18 °C. Cells were grown overnight and harvested by centrifugation at 4000 *g*. The protein was purified first by $Ni^{2+}$ affinity chromatography, then by cation exchange and finally by size-exclusion chromatography.

## Illumina sequencing (mammalian cells)

Regions of interest were amplified from genomic DNA and sequenced on an Illumina MiniSeq platform. PCR1 (98°C—2 min, 24 cycles of (98°C—15 s, 64°C—20 s, 72°C—15 s), 72°C—5 min) was performed using 200 ng gDNA, 1.25 µl of 10 µM forward and reverse primers that contain Illumina adapter sequences, 12.5 µl NEBNext UltraII Q5 Master Mix, and water to bring the total volume to 25 µl. PCR2 (98°C—2 min, 10 cycles of (98° C—15 s, 64°C—20 s, 72°C—15 s), 72°C—5 min) was done using 1 µl of unpurified PCR1 reaction mixture, 1.25 µl of 10 µM forward and reverse primers that contain unique barcode sequences, 12.5 µl NEBNext UltraII Q5 Master Mix, and water to bring the total volume to 25 µl. PCR2 products were first analyzed using 2 % agarose gel electrophoresis, and then similar amounts were pooled based on the band intensities. Pooled PCR2 products were first purified by gel extraction (Qiagen) and purified again by PCR cleanup columns (Qiagen). Concentration of final purified library was determined by Qubit (High Sensitivity DNA assay). The integrity of library was confirmed by Agilent Tapestation using Agilent High Sensitivity D1000 ScreenTape kit. The library was then sequenced on an Illumina Miniseq platform according to the manufacturer's instructions using MiniSeq Mid Output Kit (300 cycles). Sequencing reads were demultiplexed using bcl2fastq2 (Illumina) and CRISPResso2 (*Clement et al., 2019*) was used to align the reads and quantify editing efficiencies. Quantification window size was set as 30 to ensure the stringent analysis. HDR efficiency was calculated as percentage of (precise HDR reads)/(total reads).

## GUIDE-seq experiment

Two PS linkages were incorporated between the first three and the last three nucleotides in the dsODN tags. Unmodified dsODN does not contain any further modifications whereas modified dsODN contains 5′ TEG (SP9) modification (IDT). Sequencing libraries were prepared as previously described (*Tsai et al., 2015*); 1 pmol of dsODN was used. Data was processed and analyzed using the GUIDE-seq analysis software (*Tsai et al., 2015*).

## Cell culture and transfections

HEK293T cells were obtained from ATCC and were cultured in standard DMEM medium (Gibco, #11995) supplemented with 10 % fetal bovine serum (FBS) (Sigma, #F0392). HFF were maintained in DMEM medium supplemented with 20 % FBS. CHO cells (obtained from ATCC) were cultured in F-12K medium (Gibco 21127022) supplemented with 10 % FBS, and K562 cells were cultured in IMDM medium (Gibco 12440053) supplemented with 10 % FBS. Traffic Light Reporter Multi-Cas Variant 1 reporter cells were previously described (*Iyer, 2019*). Mycoplasma tests were performed prior to the experiments and all the cell lines tested negative for contamination. Electroporations were performed using the Neon transfection system (ThermoFisher). SpyCas9 was delivered either as a plasmid or as protein. For plasmid delivery of Cas9 and sgRNA, 10 µl of the mixtures containing appropriate amounts of plasmids and 100,000 cells in Neon buffer-R (ThermoFisher) were nucleofected (see manufacturer's protocol). For RNP delivery of Cas9 (IDT), GFP-to-BFP assay (20 pmol Cas9 and 25 pmol of crRNA-tracrRNA), EMX1-HEK293T (5 pmol Cas9, 10 pmol sgRNA (IDT)), EMX1-K562 (10 pmol Cas9, 20 pmol sgRNA), were mixed with buffer R. This mixture was incubated at room temperature for 30 min followed by the addition of 100,000 cells that were already resuspended in buffer R; 10 µl of this mixture containing appropriate concentrations mentioned above were then electroporated using the 10 µl Neon tips. Electroporation parameters (pulse voltage, pulse width, number of pulses) were 1150 V, 20 ms, 2 pulses for HEK293T cells, 1650 V, 10 ms, 3 pulses for CHO cells, 1400 V, 30 ms, 1 pulse for HFF cells and 1600 V, 10 ms, 3 pulses for K562 cells. Electroporated cells were harvested for fluorescence-activated cell sorting (FACS) analysis 48–72 hr post electroporation unless mentioned otherwise. Donor amounts presented in the figures represent number of moles of donor nucleofected per 100,000 cells in 10 µl mixtures (e.g., 1 pmol of donor represents 0.1 µM used for nucleofection).

## K562 GFP+ stable cell line generation

Lentiviral vector expressing eGFP was cloned using the Addgene plasmid #31482. The eGFP sequence was cloned downstream of the SFFV promoter using Gibson assembly. For lentivirus production, the lentiviral vector was co-transfected into HEK293T cells along with the packaging plasmids (Addgene 12260 and 12259) in six-well plates using TransIT-LT1 transfection reagent (Mirus Bio) as recommended by the manufacturer. After 24 hr, the medium was aspirated from the transfected cells and replaced

with fresh 1 ml of fresh DMEM media. The next day, the supernatant containing the virus from the transfected cells was collected and filtered through a 0.45 µm filter; 10 µl of the undiluted supernatant along with 2.5 µg of Polybrene was used to transduce ~1 million K562 cells in six-well plates. The transduced cells were selected using media containing 2.5 µg/ml of puromycin. Less than 20 % of the transduced cells survived, and these were then diluted into 96-well plates to select single clones. One of the K562 GFP+ clones was used for the analysis shown in this study. Cas9 was electroporated into the K562 GFP+ cells as RNP (20 pmol) with a crRNA targeting the *GFP* sequence. ssODN (66 nt) with or without end modifications was provided as donor template to convert the *GFP* coding sequence to the *BFP* coding sequence. % BFP(+) (HDR) and % GFP(-) BFP(-) (NHEJ) cells were quantified using flow cytometry.

## Flow cytometry

The electroporated cells were analyzed on a MACSQuant VYB from Miltenyi Biotec. Cells were gated first based on forward and side scattering to select 'live' cells and then for single cells. GFP-positive cells were identified using the blue laser (488 nm) and 525/50 nm filter whereas for the detection of mCherry-positive cells, yellow laser (561 nm) and 615/20 nm filter were used. BFP-positive cells were identified using the violet laser (405 nm) and 450 ± 50 nm filter. The gating strategy is shown in *Supplementary file 3*.

Southern blotting to visualize donor concatemers dsDNA donors (566 bp) were prepared using DIG labeled dUTP nucleotide mix (Sigma Aldrich #11585550910); 1.5 pmol of gel-extracted DNA was nucleofected into HEK293T (100,000) cells (Cas9 or guide RNAs were not added to the mix). Nucleofected cells were collected at various time points and pellets were frozen at –80 ° C until processed for DNA extraction. Total DNA was extracted using buffered phenol: chloroform: isoamyl alcohol and quantified using Qubit (HS-DNA). Total DNA (genomic+ exogenous) of 200 ng (0–24 hr) or about 800 ng (48 and 72 hr) was used for agarose gel (0.8%) electrophoresis. Higher amounts of DNA were loaded for the later time points to blot for roughly equal amounts of exogenous DNA and to account for the increase in total cell number over the time course; 200 pg of 566 bp and 800 pg of 13 kb DIG labeled PCR DNA were used as size markers. After electrophoresis agarose gel was treated with 0.25 N HCl (depurination) for 10 min followed by three washes with distilled water. The gel was then treated with denaturing solution (0.5 M NaOH and 1.5 M NaCl) for 20 min and another 30 min with fresh solution; followed by neutralization (2 washes 10 min each) with alkaline transfer buffer (5×SSC with 10 mM NaOH). Using alkaline transfer buffer, DNA was then transferred for 3 hr with upward capillary action onto positively charged nylon membrane (Amersham Hybond N+, RPN303B). After transfer, membrane was soaked in 5×SSC for 10 min and UV crosslinked. Blots were then processed using DIG Wash and Block buffer set (Sigma Aldrich #11585762001) according to the manufacturer's protocol. Briefly, membrane was blocked in 1 × blocking solution with maleic acid for 30 min, incubated with 1:20,000 Anti-Digoxigenin-AP, Fab fragments (Sigma Aldrich #11093274910) in 1 × blocking solution for 1 hr, washed twice with 1 × wash buffer, incubated in 1 × detection buffer and developed using CDP-star (Sigma Aldrich #12041677001).

## *C. elegans* microinjection and HDR screening

Microinjections were performed using Cas9 RNPs as previously described (*Ghanta and Mello, 2020*). dsDNA donors were generated by PCR; 25 ng/µl of unmodified or end-modified dsDNA donors were used in each injection mixture. Donors were heated and quick-cooled as previously described (*Ghanta and Mello, 2020*). Starting strain that is homozygous for 3XFLAG::GlyGly-Gly::TEV::CSR-1 allele was used to knock-in *gfp* sequence between flag and glycine-linker. crRNA (CTATAAAGACGATGACGATA NGG) with PAM site in the glycine-linker and donor DNA with arms homologous to 35 bp of 3xflag and 30 bp of 3xglycine-linker::tev flanking the gfp sequence were used. Loss of function WM702 (*eft3p::gfp(ne4807)*) reporter strain was generated in EG6070 (oxSi221 [eft-3p::GFP+ Cbr-unc-119(+)] II) strain background using CMG-48 and CMG-49 guides (see *Supplementary file 1*). *Rol-6 (su1006)* plasmid was used as co-injection marker. This marker plasmid forms episomal non-integrating extrachromosomal elements that transiently mark a subset of progeny by causing them to exhibit an easily scored Roller phenotype. Under the conditions used, high-quality injections into both gonad arms yielded 20–40 Roller progenies from each

injected animal. For each donor type entire F1 broods from four or more injected animals were scored and tabulated the total number of GFP-positive progeny and the number of GFP-positive Roller progeny.

## Zebrafish experiments

### Fish care
Fish were maintained in accordance with the protocols set by the University of Massachusetts Medical School Institutional Animal Care and Use Committee. All the injections were performed into embryos derived from in-crosses of the EK wild-type (WT) line.

### Zebrafish zygote microinjections
One cell-staged embryos were injected with 30 pg of either unmodified or end-modified donors together with 24 fmol RNP of modified Cas12a protein (Lb-2C-Cas12a) and modified crRNA (dr crRNA) per embryo as described previously (*Liu et al., 2019*), targeting 5'-end of the *hey2* coding sequence. Embryos were incubated for 24 hr post injection, genomic DNA was extracted, and libraries for amplicon sequencing were prepared. For library construction, linear amplification using a single primer containing UMI was performed first, followed by PCRs for exponential amplification and barcode stitching were performed as described previously (*Liu et al., 2019*). Quantification of the reads containing indels and precise knock-ins of Avi-tag was performed using the Python script deposited at the GitHub repository: (https://github.com/locusliu/PCR_Amplicon_target_deep_seq/blob/master/CRESA-lpp.py; copy archived at swh:1:rev:4c1209cc15d1a90a1fa8a757677e23e59ba2ed44, *Liu, 2021*). All the experiments were performed in three independent replicates except injections with unmodified donors which were performed in duplicates.

## Mouse experiments

### Strains and microinjection
All the mouse experiments were conducted according the UMMS Institute Animal Care and Use Committee (IACUC). C57BL/6 J (Stock #000664) and Swiss-Webster (Stock #SW) were obtained from Jackson Laboratory and Taconic, respectively. All the animals were maintained in a 12 hr light/dark cycle. Superovulated females were mated, and their zygotes were collected at E0.5. Male pronuclei were injected with the injection mixtures described below. Finally, zygotes were transferred to pseudo-pregnant recipients and allowed to go to term.

### Donor preparation
Using plasmids as templates and either unmodified or end-modified oligos as primers, donor sequences were PCR-amplified with Q5 polymerase (NEB). The resulting PCR products were excised from 0.8 % TAE agarose gel and purified using spin columns (Omega, #D2500). Gel-extracted DNA was further purified with 1.5 × AMPure XP (Beckman Coulter) beads according to the manufacturer's protocol and eluted in nuclease-free water. Before use in microinjection mixes, dsDNA donors were subjected to heating and cooling protocol in thermal cycler as described previously (*Ghanta and Mello, 2020*).

### Injection mixture preparation
Injections mixes were prepared with the following final concentrations: S.p. Cas9 Protein (50 ng/µl) (IDT); S.p. Cas9 mRNA (50 ng/µl) (TriLink; L-7206); sgRNA (20 ng/µl) (IDT); dsDNA donor (1 ng/µl). Cas9 protein, sgRNA, and TE (pH 7.5) were incubated at 37 °C for 20 min. This mixture was then equally split into two tubes and the following components were added to each tube: Cas9 mRNA, dsDNA donor (either unmodified or 5' 2'OMe-RNA::TEG modified), TE (pH 7.5) to bring the total volume to 50 µl. After pipetting well, the final injection mixtures were centrifuged at 14,000 *g* for 2 min and 46 µl was taken from the top (to avoid particles that may clog the needles) and transferred to fresh tubes. All the steps were performed at room temperature. Mixtures were kept on ice and directly loaded into the needles for microinjection.

## Genotyping

Tail clips of *Sox2-V5* founder animals were collected at P10, genotyped by PCR and Sanger-sequenced to confirm precise insertion. To confirm germline transmission, some of the HDR-positive F0 animals were mated with WT animals and tail clips of F1 animals were genotyped.

## Oligo sequences

Sequences of all the guide RNAs used in this study are provided in *Supplementary file 1* and sequences of all the primers used are provided in *Supplementary file 2*.

## Statistical analyses

All the statistical analyses were performed using GraphPad Prism. The type of analysis performed and the p-value information can be found in respective figures and figure legends.

## Acknowledgements

This work was funded by Howard Hughes Medical Institute (CCM), NIH R37GM058800-23 (CCM), institutional funds from University of Massachusetts Medical School (EJS and JKW), NIH UG3 TR002668 (EJS and JKW), NIH/NHLBI R35 HL140017 (NDL), and NIH/OD R21 OD030004 (NDL). We thank Darryl Conte Jr for critical reading of the manuscript, Scot Wolfe for sharing the Cas9 expression plasmid, Julia Rembetsy-Brown for maintaining *Sox2::V5* mouse colony. Biorender was used to generate some of the graphics in figures. We thank the UMass Medical School Deep Sequencing Core for their assistance with the Illumina and PacBio sequencing. Some strains were provided by the CGC, which is funded by NIH Office of Research Infrastructure Programs (P40 OD010440).

## Additional information

### Competing interests

Krishna S Ghanta, Gregoriy A Dokshin, Hassan Gneid, Erik J Sontheimer, Jonathan K Watts: Co-inventor on patent applications related to this work (Application number: 16/384, 612 ). Aamir Mir: Co-inventor on patent application related to this work (Application number: 16/384, 612 ). Craig C Mello: The authors (K.S.G, A.M, G.A.D, H.G, J.K.W, E.J.S and C.C.M) have a patent application pending related to the findings described (Application number: 16/384, 612 ). Craig C. Mello is a co-founder and Scientific Advisory Board member of CRISPR Therapeutics, and Erik J. Sontheimer is a cofounder and Scientific Advisory Board member of Intellia herapeutics.. The other authors declare that no competing interests exist.

### Funding

| Funder | Grant reference number | Author |
|---|---|---|
| Howard Hughes Medical Institute | | Craig C Mello |
| Office of Extramural Research, National Institutes of Health | R37 GM058800-23 | Craig C Mello |
| University of Massachusetts Medical School | | Erik J Sontheimer Jonathan K Watts |
| National Center for Advancing Translational Sciences | UG3 TR002668 | Erik J Sontheimer Jonathan K Watts |
| National Heart, Lung, and Blood Institute | R35 HL140017 | Nathan D Lawson |
| NIH Office of the Director | R21 OD030004 | Nathan D Lawson |

| Funder | Grant reference number | Author |
|--------|------------------------|--------|

The funders had no role in study design, data collection and interpretation, or the decision to submit the work for publication.

## Author contributions

Krishna S Ghanta, Conceptualization, Data curation, Formal analysis, Investigation, Methodology, Resources, Visualization, Writing – original draft, Writing – review and editing; Zexiang Chen, Data curation, Formal analysis, Investigation, Methodology, Resources, Writing – review and editing; Aamir Mir, Data curation, Formal analysis, Methodology; Gregoriy A Dokshin, Data curation, Methodology; Pranathi M Krishnamurthy, Hassan Gneid, Resources; Yeonsoo Yoon, Data curation, Investigation, Writing – review and editing; Judith Gallant, Ping Xu, Alireza Edraki, Methodology; Xiao-Ou Zhang, Ahmet Rasit Ozturk, Pengpeng Liu, Formal analysis, Software; Masahiro Shin, Formal analysis, Methodology, Writing – review and editing; Feston Idrizi, Methodology, Writing – review and editing; Nathan D Lawson, Formal analysis, Funding acquisition, Investigation, Methodology, Supervision, Writing – review and editing; Jaime A Rivera-Pérez, Investigation, Resources, Supervision, Writing – review and editing; Erik J Sontheimer, Funding acquisition, Project administration, Resources, Supervision, Writing – review and editing; Jonathan K Watts, Conceptualization, Funding acquisition, Investigation, Methodology, Project administration, Resources, Supervision, Writing – original draft, Writing – review and editing; Craig C Mello, Conceptualization, Formal analysis, Funding acquisition, Investigation, Project administration, Supervision, Writing – original draft, Writing – review and editing

## Author ORCIDs

Krishna S Ghanta ![ORCID] http://orcid.org/0000-0001-7502-3141
Zexiang Chen ![ORCID] http://orcid.org/0000-0002-1584-2417
Aamir Mir ![ORCID] http://orcid.org/0000-0003-0144-3914
Gregoriy A Dokshin ![ORCID] http://orcid.org/0000-0002-8780-7143
Yeonsoo Yoon ![ORCID] http://orcid.org/0000-0002-4302-9933
Masahiro Shin ![ORCID] http://orcid.org/0000-0002-0231-894X
Feston Idrizi ![ORCID] http://orcid.org/0000-0002-8035-3951
Hassan Gneid ![ORCID] http://orcid.org/0000-0001-9326-2023
Nathan D Lawson ![ORCID] http://orcid.org/0000-0001-7788-9619
Erik J Sontheimer ![ORCID] http://orcid.org/0000-0002-0881-0310
Jonathan K Watts ![ORCID] http://orcid.org/0000-0001-5706-1734
Craig C Mello ![ORCID] http://orcid.org/0000-0001-9176-6551

## Ethics

Fish were maintained in accordance with the protocols set by the University of Massachusetts Medical School Institutional Animal Care and Use Committee. All the mouse experiments were conducted according the UMMS Institute Animal Care and Use Committee (IACUC).

## Decision letter and Author response

Decision letter https://doi.org/10.7554/eLife.72216.sa1
Author response https://doi.org/10.7554/eLife.72216.sa2

# Additional files

## Supplementary files

- Supplementary file 1. Sequences of guide RNA spacers.
- Supplementary file 2. Sequences of oligos.
- Supplementary file 3. Flow cytometry analysis to determine the percentage precise and imprecise genome editing events.
- Transparent reporting form

## Data availability

Next-generation sequencing files have been deposited to Dryad (https://doi.org/10.5061/dryad.f7m0cfxwr) and NCBI Sequence Read Archive database under the BioProject accession code PRJNA768287.

The following dataset was generated:

| Author(s) | Year | Dataset title | Dataset URL | Database and Identifier |
|-----------|------|---------------|-------------|-------------------------|
| Ghanta KS | 2021 | Data from: 5′ Modifications Improve Potency and Efficacy of DNA Donors for Precision Genome Editing | https://doi.org/10.5061/dryad.f7m0cfxwr | Dryad Digital Repository, 10.5061/dryad.f7m0cfxwr |

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
