## [Decision Letter]

[Editors' note: this paper was reviewed by Review Commons.]

**Acceptance summary:**

This manuscript provides a thorough interrogation of how various modifications to ssDNA and dsDNA repair templates affect genome editing efficiency in three commonly used animals models (*Caenorhabditis elegans*, zebrafish, mice) and human cell lines. The authors convincingly demonstrate that 5' TEG modifications dramatically boost knock-in efficiency, prevent concatemer formation, and reduce imprecise editing. In zebrafish and human cells an additional methylated RNA modification (2'OMe-RNA::TEG) at the 5' end of the donor resulted in improved editing efficiency compared to TEG alone. The authors also demonstrate that a free 3'OH in donors is necessary for efficient genome editing, providing insight into the repair mechanism. This information will enable researchers to increase the efficiency of gene editing with homology templates.

---

## [Author Response]

Reviewer #1 (Evidence, reproducibility and clarity (Required)):In this manuscript, Ghanta et al. provide a thorough interrogation of how various modifications to ssDNA and dsDNA repair templates affect genome editing efficiency in three commonly used animals models (*Caenorhabditis elegans*, zebrafish, mice) and human cell lines. The authors convincingly demonstrate that 5' TEG modifications dramatically boost knock-in efficiency, prevent concatemer formation, and reduce imprecise editing. In the case of short donors, end modification did not affect the maximal level of repair, but reduced the amount of donor required to reach that maximal level. In zebrafish and human cells an additional methylated RNA modification (2'OMe-RNA::TEG) at the 5' end of the donor resulted in improved editing efficiency compared to TEG alone, whereas in *C. elegans* these two modifications performed equivalently. The authors also demonstrate that a free 3'OH in donors is necessary for efficient genome editing, providing insight into the repair mechanism. The key conclusions are rigorously tested and are convincing. The data and the methods are presented clearly, which will facilitate reproducibility and the experimental replicates and statistics were appropriate.I have two major suggestions that would further improve this work and help the wider community adopt it:1. One major limitation to this study is that the TEG and 2'OMe-RNA::TEG oligonucleotides are produced in house with an ABI 394 synthesizer followed by chemical coupling. While these methods may be routine for some labs, a large number of labs will be unable to generate these reagents which will limit the impact of this study. Indeed, these modifications were originally described in a 2018 pre-print and have not yet been widely adopted by the *C. elegans* community. I suggest that the authors repeat a few critical experiments using dsDNA donors generated with commercially produced oligos. Good candidates for such an experiment would be the Traffic Light edit from Figure 1 and csr-1 edit from Figure 4. This information could be included as a supplemental figure and should compare the performance of more widely available commercially produced, TEG and 2'OMe-RNA::TEG oligos with their in house made oligos. Information should be provided in the methods about how the oligos were ordered (scale, purity, etc.). This experiment would lower the bar and provide a benchmark for many labs to try using these modifications in their genome editing donors.

We appreciate the reviewer’s suggestions to guide the community towards commercial sources of modified donors. Indeed, most of the experiments using TEG modifications were performed with commercially sourced oligos. All the TEG donors in *C. elegans* experiments were purchased from IDT. However, all the 2′OMe-RNA:TEG modified oligos were synthesized in-house to reduce the cost burden while testing numerous different modifications at many loci. Researchers who do not have the capabilities to synthesize oligos in-house can purchase 2′OMe-RNA:TEG modified oligos from IDT. We apologize for not making this clear in the text. We have now added the details of the source, scale and purity of the commercial oligos to the supplemental methods sections.

Line 24-25: of supplemental methods “PEGylated oligos can also be purchased from commercial sources such as Integrated DNA Technologies (IDT) at 100nmol scale with simple desalting.”

2. One major difference between the *C. elegans* experiments and the experiments in other systems was that the *C. elegans* work used the heating-cooling treatment of templates that the authors recently described (Ghanta et al., 2020). As far as I can tell this heating-cooling was not done for the other organisms/cells. Have the authors tried this heating/cooling in other cells/organisms? It would be worth discussing. If not, it would be valuable to try this method with the Traffic Light edit. I don't view this experiment as essential, though it would be valuable for the community.

We thank the reviewer for the suggestion. We have added text to the Discussion section to address this concern.

Line 343 “Explorations of how modifications to the donor, both chemical and physical in nature, might alter HDR efficacy are far from complete. […] Preliminary findings suggest that melting an 800bp donor did not improve HDR efficiency in human cells (data not shown) but further studies are required to more rigorously test the impact of melting donors of greater length, and directed at more target sites, and of course to explore other physical perturbations.”

Reviewer #1 (Significance (Required)):The described donor modifications improve editing efficiency and accuracy which will significantly advance genome editing in *C. elegans*, zebrafish, mice, and human cell lines. This work builds on a 2018 pre-print from the authors and also complements and sheds light on some recent papers demonstrating that 5' biotin or ssDNA modifications boost editing efficiency (Gutierrez-Triana et al., 2018; Canaj et al., 2019). Notably these modifications also included TEG modifications, which could account for the boost in editing efficiency. This work will be of interest to any lab attempting to perform knock-ins with genome editing.Referee Cross-commentingI think that Reviewer 3's major comment is a good point and that testing the performance of the end-modified repair templates in primary cells would add a lot of value to this work.

Please see the response to reviewer 3’s comment below.

Reviewer #3 (Evidence, reproducibility and clarity (Required)):Non-viral whole-gene knock-in remains a challenge for CRISPR in many important cell types. Furthermore, the manufacture of bespoke donors at scale and of consistent quality further complicates things, and may contribute to difficulties with reproducibility that seem to be an issue in this sub-field.Here, authors utilize PCR to generate end-modified dsDNA donors that can drive efficient knock-in to immortalized cell lines and worm embryos delivered via co-electroporation with CRISPR reagents. A few corresponding studies of ssDNA donors (ssODNs) are also presented, with similar results.The experiments are well-controlled and well-conceived, particularly the use of dose-response to establish donor potency. The paper is clear and well-written. The methods are clear and appear to be readily reproduced. By targeting multiple loci and multiple cell types they provide good evidence for the robustness of the method. The results are excellent and compare favorably to other papers in a similar vein (i.e. those cited in the Discussion). In particular, they report >50% knock-in using dsDNA donors at concentrations ~30-fold less than used for short ssODNs (~0.12 µM vs. ~3 µM) which would be a significant advance in this area.The intended audience of this work are CRISPR engineers of all sorts who could use this technique to generate model cell lines, or edited model organisms. Those in cell therapy, immunology, or stem cell biology should also be interested, but data is lacking in those cell types (see below).Comment – MajorThe principal weakness of the work is over-reliance on immortalized cell lines (when electroporating), which tend to be more resilient to transfection of HDR donors than the primary (somatic) cells used for cell therapy. For example, it would be nice to know whether the increased potency of the end-modified donors also leads to declines in viability after electroporation (if the stabilized donor is more persistent), but this trade-off may only become apparent if more sensitive cells are used. This could be a limited set of in vitro experiments targeted to an endogenous locus as in Figure 2B/C/D.

We appreciate the reviewer’s suggestion to test end-modified donors in non-immortalized cells. We would argue that our abundant successes in vivo in multiple model organisms address the concern about limited benefits beyond transformed cells, and we are aware of no reason why primary cells in vitro should differ. As we note in line 340, Results from Canaj., et al. 2019 in non-transformed (iPSC) cells support our conclusions. We are indeed very interested in applying our system to primary human T cells, and our work in this area is already underway. However, this T cell work will require many distinct sets of tests and analyses, and we are therefore deferring that work to a separate manuscript in the future.

Line 339: “The third study describes the suppression of NHEJ-mediated insertions using donors with 5′-Biotin::PEG or 5′-ssDNA::PEG moieties in non-transformed cells.”

Comments – Minor– Figure 1 and elsewhere: Suggesting using "µM" of donor in the cuvette instead of "pmol" so results can be ported to other systems with different volumes (Lonza, MaxCyte, etc).

We apologize for the confusion this may have caused. We have added µM conversion details to the methods section to be able to compare with other transfection systems.

Line100 of supplemental methods: “Donor amounts presented in the figures represent number of moles of donor nucleofected per 100, 000 cells in 10 µl mixtures (e.g., 1 pmol of donor represents 0.1µM used for nucleofection).”

– Figure S1: It is intriguing that attachment of NLS peptides does not increase HDR. Use of PNA-RNA annealing is a good clean test of this hypothesis.

We thank the reviewer for this suggestion. Indeed, our experiments were performed by annealing PNA to RNA tails. We have modified the figure legend to convey this message and avoid any confusion.

“PNA::NLS and 2′OMe::TEG modified donors were mixed at 4:1 molar ratio, heated to 94°C and cooled to anneal PNA to 2′OMe-RNA.”

– Figure 2: Baseline (unmod) donor HDR rates vary from locus to locus, as does the enhancement with the 5' mods. Interesting! Can authors comment further on why rates may vary from one site to another? Why would fold HDR enhancement vary from one site to the next as well?

We thank the reviewer for the insightful comments. We agree that HDR efficiency varies from locus to locus. We added the following text to the Results section to address this comment.

Line 142-145: “HDR efficiencies obtained with unmodified donors vary from locus to locus. […] These factors may also influence the fold change increase by end-modified donors.”

– Figure 6A: A bit more explanation (in results or in legend) might be appropriate. How does HDR allow a protected donor to concatenate at all? This is important because at first the presence of concatemers when using the protected donors might be confusing.

We apologize for the confusion. We realized that the DNA cartoon in figure 6A can be misunderstood as end-modified DNA. We showed unmodified donors with head and tail representation. We modified the drawing to avoid confusion and explained in the figure legend that the unmodified donors are shown in the figure.

Figure legend has been modified to reflect the changes as, “(A) Model for mechanisms of concatemer formation for unmodified donors is shown”.

– Figure 6B: Authors note a "significant" difference in concatenation in situ with the modified donors. I agree this seems clear from the figure but some quantification (densitometry) might help bolster their case.

We thank the reviewer for this suggestion. We agree the difference is clear but the use of the word “significant” is not appropriate. The affect is obvious without densitometry and so we have simply removed the word “significant” from lines 258 and 280.

Line 258: “These results suggest that the 5′-TEG modification suppresses concatemer formation.”

Line 280” “Modifying the ends of the donors suppressed concatemer formation and reduced random integration of short dsDNA at chromosomal DSBs.”

– Figures 6C + 6D (GUIDE-seq): I appreciate the clever use of the GUIDE-seq method. It seems authors are using the published probe sequence as dsDNA, and then comparing modified dsDNA to unmodified, but this is not clear enough in the text.

We apologize for the confusion. We have added the following text to make it clear that we used the published probe sequence.

Line 263-265 “ Using the previously described dsDNA probe either with or without the TEG modifications, we targeted the *ARHGEF9* locus that was previously characterized for off-target editing”

– Discussion: Authors introduce the important concept of "dose-limiting" toxicity but don't present any viability data (nor data from primary cells wherein this is important, see above). I suggest authors focus on the data actually present, or include more toxicity data in the revised manuscript.

We agree with the reviewer that dose-limiting tox is an important concept. We noted in the text that increased cell death was correlated with the plateau in efficacy. We feel that the possibility that efficacy and toxicity of the modified donor are both increased is too important to leave from the discussion. Therefore, we have opted to leave in the manuscript this one sentence, prefaced with the condition that further study is needed, that addresses dose-limiting toxicity:

Line 289 “This difference requires further study but could be explained if shorter donors and longer DNA donors experience different dose-limiting barriers. For example, the dose-limiting toxicity of ssODNs could be driven by total number of free DNA ends per cell, while longer molecules could encounter dose-limiting toxicity driven by total DNA mass.”

Discussion line 322: Confirm whether "dose" here means "concentration of donor in cuvette" (see first comment).

We apologize for not clarifying this. As explained in response to the first comment we have added text to the methods section to clarify µM and Pmol conversion. We modified the text to avoid confusion.

Line 328: “In previous studies, fluorescent and amine modifications to the 5′ and 3′ termini of ssODN donors did not improve HDR efficacy over unmodified donors. [,,,] However, these studies were performed using much higher concentrations than the optimal concentrations for modified donors determined here.”

Reviewer #3 (Significance (Required)):This represents an excellent start to tacking a large problem: achieving efficient non-viral knock-in with CRISPR. More work will be needed to prove that it works in systems not utilized here.This work is line with earlier work in this area, most of which is cited and discussed in the Discussion section.The intended audience of this work are CRISPR engineers of all sorts who could use this technique to generate model cell lines, or edited model organisms. Those in cell therapy, immunology, or stem cell biology should also be interested, but data is lacking in those cell types.

We thank the reviewer for insightful comments and suggestions.